

# Atmospheric oxidation of new 'green' solvents part II: methyl pivalate and pinacolone

Caterina Mapelli[1,2], James K. Donnelly[1], Úna E. Hogan[1,3], Andrew R. Rickard[1,4], Abbie T. Robinson[1], Fergal Byrne[1,5], C. Rob McElroy[1,6], Basile F. E. Curchod[7], Daniel Hollas[7], and Terry J. Dillon[1]

[1] Department of Chemistry, University of York, York, YO10 5DD, UK
[2]Now at the Institute of Atmospheric Sciences and Climate, National Research Council, Lecce, 73100, Italy
[3]Now at Department of Chemistry, University of Waterloo, Waterloo, ON N2L 3G1, Canada
[4]National Centre for Atmospheric Science, Wolfson Atmospheric Chemistry Laboratories, Department of Chemistry, University of York, YO10 5DD, UK
[5]Department of Chemistry, Maynooth University, Maynooth, Co. Kildare, W23 F2H6, Ireland
[6]School of Chemistry, University of Lincoln, Lincoln, LN6 7TS, UK
[7]School of Chemistry, University of Bristol, Bristol, BS8 1QU, UK

*Correspondence to*: Terry J. Dillon (*terry.dillon@york.ac.uk*)

**Abstract.** Lab-based experimental and computational methods were used to study the atmospheric degradation of two promising "green" solvents: pinacolone, $(CH_3)_3CC(O)CH_3$ and methyl pivalate, $(CH_3)_3CC(O)OCH_3$. Pulsed laser photolysis coupled to pulsed laser induced fluorescence was used to determine absolute rate coefficients (in $10^{-12}$ cm$^3$ molecule$^{-1}$ s$^{-1}$) of $k_1(297 \text{ K}) = (1.2 \pm 0.2)$ for OH + $(CH_3)_3CC(O)CH_3$ (R$_1$) and $k_2(297 \text{ K}) = (1.3 \pm 0.3)$ for OH + $(CH_3)_3CC(O)OCH_3$ (R$_2$), in good agreement with one previous experimental study. Rate coefficients for both reactions were found to increase at elevated temperature, with $k_1(T)$ adequately described by $k_1(297 - 485 \text{ K}) = 2.1 \times 10^{-12} \exp(-200/T)$ cm$^3$ molecule$^{-1}$ s$^{-1}$. $k_2(T)$ exhibited more complex behaviour, with a local minimum at around 300 K. In the course of this work, $k_3(295 - 450 \text{ K})$ for the well-characterised reaction OH + $C_2H_5OH$ (ethanol, R$_3$) were obtained, in satisfactory agreement with the evaluated literature.

UV-vis. spectroscopy experiments and computational calculations were used to explore $(CH_3)_3CC(O)CH_3$ photolysis (R$_4$). Absorption cross sections for $(CH_3)_3CC(O)CH_3$, $\sigma_4(\lambda)$, in the actinic region were larger and the maximum was red-shifted compared to estimates used in current state-of-science models. As a consequence, we note that photolysis (R$_4$) is likely the dominant pathway for removal of $(CH_3)_3CC(O)CH_3$ from the troposphere. Nonetheless, large uncertainties remain as quantum yields $\phi_4(\lambda)$ remain unmeasured. Lifetime estimates based upon (R$_1$) and (R$_4$) span the range 2 – 9 days and are consequently associated with a poorly constrained Photochemical Ozone Creation Potential estimate (POCP$_E$). In accord with previous studies, $(CH_3)_3CC(O)OCH_3$ did not absorb in the actinic region, allowing for straightforward calculation of an atmospheric lifetime of ≈ 9 days and a small POCP$_E$ ≈ 11.



## 1 Introduction

Solvents are of fundamental importance to the chemical industry, however many common solvents are sourced from the petrochemical industry and are classed as hazardous pollutants; their usage should be phased-out as we move towards a net-zero carbon economy (Clarke et al., 2018; Byrne et al., 2016; Winterton, 2021). Solvents were the dominant source of volatile organic compound (VOC) emissions in England in 2019 (Jones et al., 2021), and so can have a potentially important impact on air quality. A solvent of particular concern is the monoaromatic species toluene ($C_6H_5CH_3$) (Calvert et al., 2002), known to pose significant human health and environmental hazards (Byrne et al., 2018a) and classified as a significant air pollutant (Montero-Montoya et al., 2018; Zhang et al., 2019). This work follows on from Mapelli et al. (2022), where the atmospheric oxidation of 2,2,5,5-tetramethyloxolane (TMO) (Byrne et al., 2017), was studied in detail ('part I'). Byrne et al. (2018b) similarly identified pinacolone (3,3-dimethyl-2-butanone, $(CH_3)_3CC(O)CH_3$), henceforth PCO, and methyl pivalate (methyl 2,2-dimethylpropanoate, $(CH_3)_3CC(O)OCH_3$), henceforth MPA as potential replacements for hazardous volatile non-polar solvents such as toluene. Advantages include low toxicity, sustainable (biomass) sourcing and good solvation characteristics. However, whilst the atmospheric chemistry of toluene is well known and its deleterious air quality impacts have been quantified, the atmospheric breakdown of PCO or of MPA was seldom studied. No air quality considerations were included in their selection as "green" solvents (Byrne et al., 2018a).

Industrially, PCO is produced for use in fungicides, herbicides, and pesticides (Liu et al., 2022). In addition to direct emission from industry, the Master Chemical Mechanism (MCM, mcm.york.ac.uk) (Saunders et al., 2003b; Jenkin et al., 1997), a benchmark detailed description of atmospheric chemistry, reports PCO as a product of gas-phase oxidation of both 2,2-dimethylbutane and 3,3-dimethyl-2-butanol. An important removal process from the troposphere for virtually all VOC, including PCO, is via reaction with the hydroxyl radical, OH ($R_1$, see Fig. 1).

$$OH + PCO \quad \rightarrow \quad (products) \tag{R_1}$$

The ambient temperature rate coefficient of ($R_1$) was measured by Wallington and Kurylo (1987a) as $k_1(296\ K) = (1.21 \pm 0.05) \times 10^{-12}$ cm$^3$ molecule$^{-1}$ s$^{-1}$; this value is used in the MCM, where ($R_1$) is described as proceeding exclusively via beta-$CH_2$-H abstraction (R1b) (Saunders et al., 2003b; Jenkin et al., 1997). An alternative breakdown process for carbonyl containing VOC in the atmosphere is photolysis ($R_4$), where the C=O function may absorb abundant UV light at $\lambda > 290$ nm via the n $\rightarrow$ $\pi*$ electronic transition.

$$PCO + h\nu\ (\lambda > 290\ nm) \rightarrow (products) \tag{R_4}$$

PCO is known to have an absorption maximum at 287 nm in cyclohexane solution (Chimichi and Mealli, 1992) and at 278 nm in water (Pocker et al., 1988) though quantitative absorption cross sections were not reported. In the absence of quantitative cross-sections or quantum yields, the MCM represents PCO photodissociation ($R_4$) and subsequent rapid reactions with $O_2$ as exclusively producing $(CH_3)_3CO_2 + CH_3C(O)O_2$, and the photolysis rate parameters of PCO (and seven other ketones) via parameters measured for photolysis ($R_5$) of methyl ethyl ketone (butanone, $CH_3C(O)C_2H_6$, henceforth MEK).

$$MEK + h\nu\ (\lambda > 290\ nm) \rightarrow (products) \tag{R_5}$$





With a similar branched chemical structure, and different functionality, the ester MPA was also selected by Byrne et al. (2018b), as a bio-based replacement for problematic hydrocarbon solvents. A wide range of industries make use of MPA, including as a solvent with low ozone formation potential. Wallington et al. (2001) have studied the principal atmospheric removal reaction for MPA ($R_2$), reporting $k_2$(298 K) = (1.20 ± 0.03) × 10$^{-12}$ cm$^3$ molecule$^{-1}$ s$^{-1}$, and complex, non-Arrhenius behaviour for $k_2$(250 - 370 K).

70            OH + MPA        → (products)                                                    ($R_2$)

The UV absorption spectrum of MPA was also measured by Wallington et al. (2001), who reported an absorption maximum at around 210 nm but no absorption at wavelengths ($\lambda > 290$ nm) where UV light is abundant in the troposphere.



**Figure 1. Proposed pathways for PCO + OH (R1, left), where the H abstraction may proceed at the alpha CH$_2$-H (R1a) or beta CH$_2$-**
**H (R1b, as proposed in the MCM). Similarly, two different sites for H-abstraction are available for the reaction MPA + OH (R2, right). In the atmosphere, these radical products will react rapidly with oxygen to form peroxy radicals (RO$_2$).**

Reactions of OH with oxygenated volatile organic compounds (OVOC) such as PCO and MPA are notoriously complex, with several examples displaying bimodal non-Arrhenius kinetic behaviour at atmospherically relevant temperatures (Calvert et al.,
2011; Wollenhaupt et al., 2000). The lack of photochemical data for degradation of PCO and MPA in the troposphere, and OVOCs in general, is therefore concerning. Rate coefficients $k_1$ and $k_2$ were each reported just once (Wallington and Kurylo, 1987a; Wallington et al., 2001; Mapelli et al., 2022). Photolysis ($R_4$) is potentially the dominant loss process for PCO, yet the state of science model MCM (Master Chemical Mechanism) represents $R_4$ using absorption cross-sections and quantum yields for a surrogate ketone species, MEK (Saunders et al., 2003a). This paucity of data, together with the recent increase in interest
around "green" solvents were prime motivations for this work, which aims to improve our knowledge of the atmospheric chemistry of the OVOC species PCO and MPA. Direct, absolute kinetic methods were used to determine temperature dependent rate coefficients for reaction of PCO and MPA with OH. Absorption cross sections for PCO were obtained from both laboratory and computational experiments. These fundamental photochemical parameters were then used to calculate lifetimes for removal of PCO and MPA from the troposphere. Finally, photochemical ozone creation potentials (POCP$_E$) were
estimated according to the method proposed by Jenkin et al. (2017), allowing for air quality impact assessments of these molecules for use as "green" solvents.



## 2 Experimental and computational methods

Two distinct sets of experiments were used to examine solvent breakdown reactions. In section 2.1, a brief description of the
University of York pulsed laser apparatus is provided. This setup, used to obtain absolute rate coefficients $k_1$(295 – 485 K) and
$k_2$(295 – 485 K) was described more fully in a recent publication (Mapelli et al., 2022). Section 2.2 describes UV-vis.
measurements of solution-phase absorption cross-sections, $\sigma(\lambda)$ for PCO and for MEK, a compound that has measured gas-
phase absorption cross sections and that to date, has been used to represent PCO photolysis (as well as a range of structurally
similar ketone species) in models of atmospheric chemistry (Jenkin et al., 2003; Saunders et al., 2003a). The computational
methods used to for in-silico $\sigma(\lambda)$ calculations are presented in section 2.3.

### 2.1 Pulsed Laser Photolysis – Laser Induced Fluorescence (PLP-LIF) experiments

The absolute kinetic technique of pulsed laser photolysis (PLP), coupled to direct pulsed laser induced fluorescence (LIF)
detection of OH, was used to determine rate coefficients $k_1$(295 – 485 K) and $k_2$(295 – 485 K). This apparatus was described
in detail in a previous publication (Mapelli et al., 2022). Briefly, output from four mass flow controllers was mixed prior to
entering a 400 $cm^3$ Pyrex reactor equipped with thermocouples and capacitance manometers. Output from these calibrated
analogue devices was digitised and passed to a PC to regulate and log gas temperature, pressure and flow rate. This data was
subsequently used for manometric calculation of reagent concentrations; rate coefficient determinations in this work relied
critically on accurate knowledge of [OVOC], here estimated to a precision of ±15%.

The fourth harmonic output ($E$ = 20 mJ $cm^{-2}$ / pulse) from a 10 Hz Nd:YAG laser was directed into the reactor via a quartz
Brewster window and used to generate OH via 266 nm PLP of $H_2O_2$ ($R_6$).

$$H_2O_2 + h\nu \rightarrow 2OH \tag{$R_6$}$$

Kinetic experiments conducted in the absence of OVOC, and manometric estimates both indicated that $[H_2O_2] \approx 10^{14}$ molecule
$cm^{-3}$ in all experiments. Under these conditions, an estimated $[OH] \approx 10^{11}$ molecule $cm^{-3}$ was generated ($R_6$). As such, pseudo
first order conditions of [OVOC] >> [OH] applied throughout, save in experiments conducted with [OVOC] = 0. The 282 nm
output from a frequency-doubled dye laser (Rhodamine-6G; pumped at 532 nm) was directed through a second quartz Brewster
window, co-linear but counter-propagating to the PLP laser. This tuneable laser light was used to pump the $Q_{11}$ transition of
$A^2\Sigma^+(v = 1) \leftarrow X^2\Pi(v = 0)$ at 281.997 nm for direct, off-resonant LIF detection of OH. Fluorescence decay signals $S$ were
monoexponential, allowing for analysis via fitting with equation (Eq1):

$$S(t) = S_0 \exp(-Bt) \tag{Eq1}$$

where $S_0$ refers to the LIF signal at $t = 0$ and is proportional to the initial concentration of OH generated by ($R_6$) and $B$ represents
the pseudo-first order rate coefficient.





## 2.2 Absorption cross sections, $\sigma(\lambda)$, via UV-vis. experiments

UV-vis. studies were carried out using a double beam Shimadzu UV-2600 spectrometer over the wavelength range 250-400 nm at $T = (298 \pm 1)$ K and with a resolution of 1 nm. The spectra were recorded in Hellma Analytics quartz cuvettes with a

path length of 10 mm or 2 mm. As no suitable apparatus was available at University of York for gas-phase $\sigma_4(\lambda)$ determinations, all spectra recorded were solution phase, diluted in cyclohexane ($c$-$C_6H_{12}$). This choice was based on the work of Nakashima et al. (1982), who studied the solvent effect on the absorption spectra of ketones. Spectra in perfluorohexane ($C_6F_{14}$) were reported as *quasi*-vapour spectra. In this work, use of $C_6F_{14}$ was avoided for environmental reasons (Tsai, 2009); hence the similarly non-polar and weakly polarisable cyclohexane was chosen to imitate $C_6F_{14}$. Solutions (0.03 M – 0.3 M)

were prepared using volumetric flasks and micropipettes, with the concentration uncertainty estimated at around 10%.

**Chemicals:** $N_2 > 99.9999\%$ was obtained directly from $N_2(l)$ boil-off; $O_2$ (99.995%, BOC) was used as supplied; $H_2O_2$ (JT Baker, 60% in $H_2O$) was prepped to an estimated (vapour pressure) mixing ratio of $> 90\%$ by continuous flow of $N_2$ through the liquid to remove the more volatile $H_2O$ component then supplied via a bubbler maintained at $T = 273$ K and close to reactor

pressure; MPA (98%, Sigma-Aldrich), PCO (97%, Sigma-Aldrich) and $C_2H_5OH$ (99%, Sigma-Aldrich) were subject to repeated freeze-pump-thaw cycles at $T = 77$ K prior to dilution with $N_2$ (mixing ratios $\approx 0.5\%$) in 12 dm$^3$ pyrex bulbs for storage and supply. For UV-vis. experiments reagents MPA, PCO, $c$-$C_6H_{12}$ (99% Sigma-Aldrich) and MEK (>99.7% Sigma-Aldrich) were used without further purification.

## 2.3 Computational methods used to determine $\sigma(\lambda)$

Ground-state minimum-energy geometries for PCO and MEK were located with density functional theory (DFT) calculations employing the ωB97XD (Chai and Head-Gordon, 2008) exchange-correlation functional and the def2-TZVP basis set (Weigend and Ahlrichs, 2005), both in gas phase and using an implicit solvent within the IEF-PCM framework (using parameters for cyclohexane). Each stationary point located was confirmed to be a minimum by performing a frequency calculation at the very same level of theory. Two different conformers were located for MEK (in gas phase and with the implicit

solvent model). The free energy difference between the two conformers is 0.071 eV in gas phase and 0.059 eV in solution (corresponding to a Boltzmann population ratio of 0.942:0.058 in gas phase and 0.909:0.091 in solvent at 298.15 K). Photoabsorption cross-sections were calculated following the workflow described by (Prlj et al., 2022). In short, a Wigner function for uncoupled harmonic oscillator was created from the frequencies (discarding the lowest-energy mode, < 100 cm$^{-1}$) and equilibrium geometry for each molecule and each conformer, from which 1000 different geometries were sampled. The

nuclear ensemble approach (Crespo-Otero and Barbatti, 2012) was then employed to construct the photoabsorption cross-sections for each molecule (and conformer) from the transition energy and oscillator strength ($S_1 \leftarrow S_0$) of each sampled geometry obtained with linear-response time-dependent DFT (LR-TDDFT) within the Tamm-Dancoff approximation (TDA), using the same exchange-correlation functional and basis set as earlier (ωB97XD and def2-TZVP), with and without a LR-





PCM (Cammi et al., 2000). This level of electronic-structure theory was benchmarked against EOM-CCSD/aug-cc-pVDZ and

was found to produce reliable excitation energies and oscillator strengths in earlier studies (Sarkar et al., 2021). LR-TDDFT/TDA/ωB97XD/def2-TZVP gives a vertical excitation (oscillator strength) towards $S_1$ of 4.61 eV (0.0000) for MEK and 4.49 eV (0.0002) for PCO, while EOM-CCSD/aug-cc-pVDZ places these transitions at 4.62 eV (0.0000) for MEK and 4.50 eV (0.0001) for PCO. The final photoabsorption cross-section of MEK (in gas phase and in solution) was obtained by averaging the photoabsorption cross-section of each conformer, weighted by the corresponding Boltzmann population. The

NEA calculations were performed with Newton-X version 2.4 (Barbatti, 2013; Barbatti et al., 2014), with electronic-structure information provided by Gaussian16 Revision B.01 (Frisch et al., 2016).

## 3 Results and discussion

Results from absolute kinetic determinations of $k_1$(295 – 485 K) and $k_2$(295 – 485 K) are presented in section 3.1. Absorption

cross-section data determined in this work are presented in sections 3.2 (experimental UV-vis. results) and 3.3 (results from computational studies). In section 3.4, these results are discussed in full, with an emphasis on the relative importance of photolysis vs. the OH reaction route and on outstanding uncertainties.

### 3.1 Absolute determinations of $k_1$(295 – 485 K) and $k_2$(295 – 485 K).

Figure 2 shows a typical exponential LIF signal decay, obtained from a $R_1$ experiment conducted at $T = 460$ K under pseudo-

first order conditions of [PCO] > 1000 × [OH]. The insert on Figure 2 summarises results from 12 such experiments, with pseudo-first order rate parameter, $B$ (Eq1) plot vs. corresponding [PCO]. The slope from a linear fit to this dataset yields the second order rate constant $k_1$(460 K).





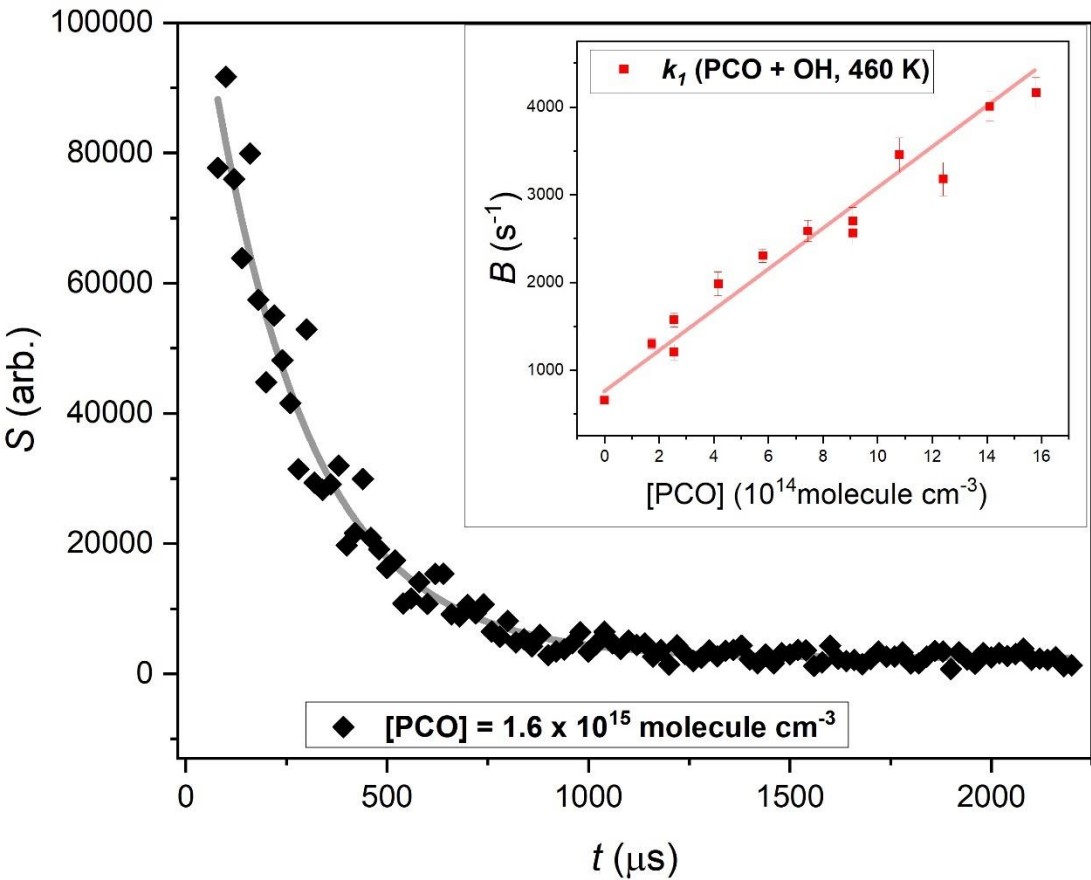

**Figure 2: Exemplary exponential decay of OH LIF signal in the presence of excess [(CH₃)₃CC(O)CH₃] = 1.6×10¹⁵ molecule cm⁻³, fit**
**with (Eq. 1) to yield pseudo first-order rate coefficient $B$ = (4090 ± 153) s⁻¹. Inset shows bimolecular plot used to determine $k_1$(460 K)**
**= (2.3 ± 0.1) × 10⁻¹² cm³ molecule⁻¹ s⁻¹.**

Similar experiments were conducted at different temperatures, pressures and in the presence / absence of $O_2$. Results from all

such $k_1(T)$ determinations are listed in Table 1 and displayed in Arrhenius format in Figure 3. A weighted average of the three

values at close to ambient temperature yields $k_1$(297 K) = (1.23 ± 0.09) × 10⁻¹² cm³ molecule⁻¹ s⁻¹ in good agreement with the

solitary previous determination of $k_1$(296 K) = (1.21 ± 0.05) × 10⁻¹² cm³ molecule⁻¹ s⁻¹ (Wallington and Kurylo, 1987a). Over

the range of temperatures investigated in this work these results were adequately represented by a two-parameter Arrhenius

equation, as $k_1$(297 – 485 K) = 2.14×10⁻¹² exp(-200/$T$) cm³ molecule⁻¹ s⁻¹. Also displayed in Fig. 3 are results from this work

on the well-characterised reaction between ethanol (CH₃CH₂OH) and OH (R₃), many of which were obtained in back-to-back

experiments alongside those on (R₁).



$$CH_3CH_2OH + OH \rightarrow products \tag{R_3}$$

The satisfactory agreement between the $k_3$(295 – 447 K) determinations here and the extensive literature dataset (Dillon et al., 2005; Carr et al., 2008; Wallington and Kurylo, 1987b; Hess and Tully, 1988; Jiménez et al., 2003) lend confidence to the $k_1(T)$ results from this work.


**Figure 3 - Arrhenius plot for the reaction of PCO + OH (R1): filled black square datapoints represent $k_1(T)$ determinations from this work using $N_2$ bath gas; open black squares using air; the filled blue circle represents the solitary determination from Wallington et al. (1987); the dashed black line an Arrhenius fit to all $k_1(T)$ results to yield $k_1$(296 or 297 – 485 K) = 2.14×10$^{-12}$ exp(-200/$T$) cm$^3$ molecule$^{-1}$ s$^{-1}$; the solid black line represents $k_1(T)$ calculated using the SAR proposed by Jenkin et al. (2018). Also displayed are $k_3(T)$ from this work (black triangles) and from literature (Dillon et al., 2005; Hess and Tully, 1989; Carr et al., 2008; Wallington and Kurylo, 1987b; Jiménez et al., 2003) together with the corresponding IUPAC evaluation (Atkinson et al., 2006) leading to the three-parameter expression $k_3(T) = 6.70 \times 10^{-18} T^2$ exp(511/$T$) cm$^3$ molecule$^{-1}$ s$^{-1}$, here represented with a red dashed line.**





A similar series of experiments was undertaken to determine $k_2$(295 - 485 K), results from which are presented in Table 1 and

depicted in Arrhenius format on Figure 4. Control experiments were conducted at different pressures and in both $N_2$ and air

bath gas to ensure the reactivity was not affected by these factors; again, many experiments were conducted back-to-back with

equivalent $k_6$ determinations (see above). A weighted mean of the three, room temperature determinations led to a value of

$k_2$(297 K) = (1.30 ± 0.01) × $10^{-12}$ $cm^3$ molecule$^{-1}$ s$^{-1}$. This result agrees well with the ambient temperature value reported by

(Wallington et al., 2001) of $k_2$(296 K) = (1.20 ± 0.03) × $10^{-12}$ $cm^3$ molecule$^{-1}$ s$^{-1}$ when considering potential systematic

uncertainties, notably in [MPA] (see section 3.4 below). As is evident from Figure 4, results from this work confirmed the

complex non-Arrhenius behaviour reported by Wallington et al. (2001), revealing a local minimum in $k_2(T)$ close to ambient

temperature. Such non-Arrhenius behaviour may be indicative of a change of mechanism, with direct hydrogen abstraction

dominating at high temperatures, whilst pathways via hydrogen bonded pre-reaction complexes play an increasingly important

role at lower temperatures. A similar trend was observed for reactions of OH with other OVOC (Wollenhaupt et al., 2000;

Vasvári et al., 2001). As displayed by the black dashed line on Figure 4, the full set of $k_2(T)$ results from this work and from

Wallington et al. can be adequately represented by a three-parameter equation $k_2$(250 – 485 K) = [50.3 exp(-1740/$T$) + 0.43

exp(276/$T$)]×$10^{-12}$ $cm^3$ molecule$^{-1}$ s$^{-1}$.




**Table 1 – absolute determinations of $k_1$, $k_2$ and $k_3$ from this work**

| Reaction | T (K) | P (Torr) [a] | [OVOC] [c] | n [d] | k [e] |
|---|---|---|---|---|---|
| (R1) | 297 | 46 | 11 - 112 | 10 | 1.06 ± 0.06 |
| (R1) | 297 | 46[a] | 5.2 - 92 | 10 | 1.31 ± 0.06 |
| (R1) | 298 | 47 | 11.0 - 60 | 14 | 1.33 ± 0.06 |
| (R1) | 341 | 60 | 2.3 - 21 | 12 | 1.39 ± 0.11 |
| (R1) | 343 | 52 | 1.3 - 20 | 12 | 1.63 ± 0.15 |
| (R1) | 343 | 52 | 1.3 - 20 | 12 | 1.5 ± 0.10 |
| (R1) | 372 | 61 | 2.4 - 37.9 | 15 | 1.76 ± 0.09 |
| (R1) | 400 | 68 | 2.6 - 28.4 | 12 | 1.82 ± 0.10 |
| (R1) | 424 | 63 | 2.4 - 22.5 | 12 | 1.59 ± 0.06 |
| (R1) | 424 | 63[a] | 2.4 - 22.5 | 12 | 1.8 ± 0.05 |
| (R1) | 460 | 68 | 2.0 - 22.5 | 12 | 2.40 ± 0.11 |
| (R1) | 462 | 52 | 1.7 - 16 | 12 | 2.33 ± 0.13 |
| (R1) | 485 | 68 | 1.9 - 21 | 11 | 2.75 ± 0.14 |
| (R2) | 295 | 61.5 | 2.0 - 18 | 10 | 1.35 ± 0.13 |
| (R2) | 297 | 80 | 3.5 - 45 | 13 | 1.29 ± 0.04 |
| (R2) | 297 | 108 | 4.6 - 35 | 16 | 1.34 ± 0.06 |
| (R2) | 297 | 80[b] | 3.5 - 45 | 13 | 1.29 ± 0.03 |
| (R2) | 340 | 59 | 1.8 - 16 | 17 | 1.15 ± 0.08 |
| (R2) | 372 | 61 | 3.3 - 25 | 15 | 1.60 ± 0.06 |
| (R2) | 400 | 68 | 1.7 - 18.4 | 12 | 1.72 ± 0.10 |
| (R2) | 424 | 63[b] | 1.3 - 20 | 15 | 1.47 ± 0.05 |
| (R2) | 424 | 63 | 1.3 - 17 | 15 | 1.63 ± 0.06 |
| (R2) | 460 | 52 | 1.4 - 12 | 12 | 2.1 ± 0.2 |
| (R2) | 485 | 69 | 1.4 - 15.5 | 12 | 2.32 ± 0.14 |
| (R3) | 295 | 57 | 2.3 - 32.7 | 15 | 3.12 ± 0.11 |
| (R3) | 297 | 49 | 4.8 - 29.4 | 8 | 3.2 ± 0.2 |
| (R3) | 297 | 49 | 4.8 - 29.4 | 8 | 3.0 ± 0.2 |
| (R3) | 345 | 71 | 3.3 - 23.4 | 10 | 3.66 ± 0.11 |
| (R3) | 408 | 70 | 2.8 - 16.1 | 12 | 4.04 ± 0.14 |
| (R3) | 447 | 70 | 2.5 - 15 | 10 | 4.48 ± 0.11 |

**Key:** [a] = pressure of $N_2$ bath gas except [b] = air bath gas; [c] = units for [OVOC] were $10^{14}$ molecule cm$^{-3}$; [d] = number of different [OVOC] in bimolecular plot; [e] = k values in units of $10^{-12}$ cm$^3$ molecule$^{-1}$ s$^{-1}$.



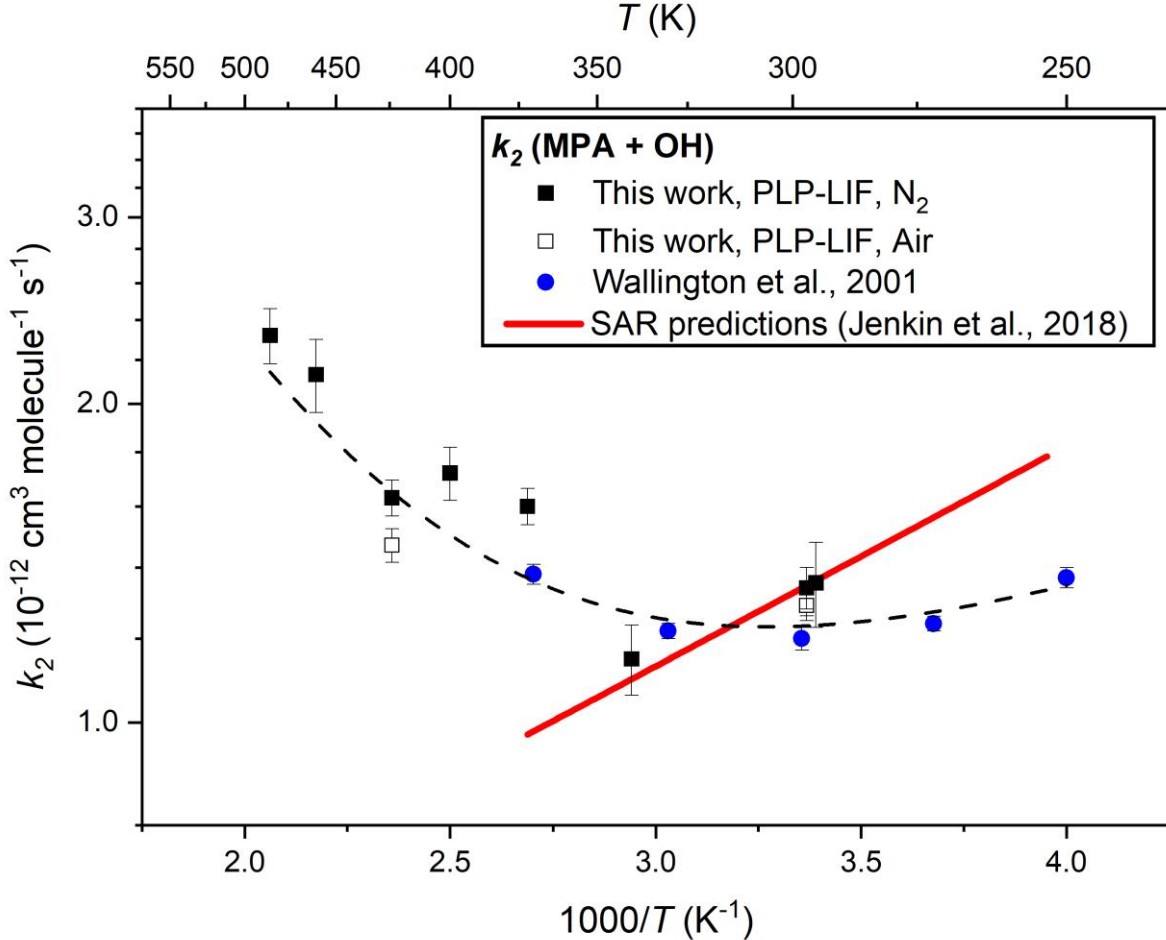

**Figure 4: Arrhenius plot for the reaction of MPA with OH (R2): filled black square datapoints represent $k_2(T)$ determinations from this work using $N_2$ bath gas; open black squares using air; the filled blue circles indicate literature data from Wallington et al. (1987); the black dashed line represents a fit to all experimental data to yield $k_2(T) = 5.0 \times 10^{-11} \exp(-1743/T) + 4.3 \times 10^{-13} \exp(276/T)$. The solid red line represents $k_2(T)$ calculated from the SAR proposed by Jenkin et al. (2018).**

### 3.2 UV-vis. cross-sections $\sigma(\lambda)$ determined in the laboratory

Figure 5 displays the solution-phase spectrum for PCO (in cyclohexane) obtained using the methods detailed in section 2.2 above. Previous spectra of PCO over the wavelength range of interest were not available, except for the absorption maximum recorded for PCO in water (Pocker et al., 1988) and in cyclohexane (Chimichi and Mealli, 1992) at 278 nm and 287 nm respectively. That such dilute, solution phase spectra can yield useful results is demonstrated by equivalent spectra for MEK. Cross-section values similarly obtained in this work, dilute in cyclohexane are displayed on Fig. 5 alongside the gas-phase spectrum reported by Martinez et al. (1992) recommended by IUPAC (Atkinson et al., 2006) and confirmed by Brewer et al.



(2019) in more recent studies. The solution phase determinations consistently overestimate reported gas phase values (e.g. at
maximum value, $\sigma_{(278\,nm)} = 6.2 \times 10^{-20}$ cm$^2$ molecule$^{-1}$ from this work vs. $\sigma_{(278\,nm)} = 5.8 \times 10^{-20}$ cm$^2$ molecule$^{-1}$ in the gas-phase,
(Martinez et al., 1992) see supplemental information). Whilst this was within the 10% uncertainty estimated for this work,
observing the spectra over at least three repeats from different solutions the overestimation appears to be systematic (see SI)
and was also identified *in silico* (see section 3.3 below). The insert to Fig. 5 displays Beer-Lambert plots for data recorded in
this work at 290 nm, yielding the molar absorption coefficients $\varepsilon_{4(290\,nm)} = (20.6 \pm 0.2)$ cm$^{-1}$ M$^{-1}$ for PCO and $\varepsilon_{5(290\,nm)} = (16.4$
$\pm 0.2) \times 10^{-20}$ cm$^{-1}$ M$^{-1}$ for MEK. These coefficients can then be converted to the absorption cross sections, leading to $\sigma_{(290nm)}$
$= (8.0 \pm 0.8) \times 10^{-20}$ cm$^2$ molecule$^{-1}$ for PCO and $\sigma_{(290nm)} = (6.0 \pm 0.6) \times 10^{-20}$ cm$^2$ molecule$^{-1}$ for MEK. The UV spectra of PCO
determined in this work were clearly characterised by larger cross sections in the actinic region (see Fig.6) and by a
significantly red-shifted local maximum in absorption (286 nm vs. 278 nm) when compared to MEK. These observations will
therefore have significant implications when revising estimates for photolysis rates of PCO, as more light is available at longer
wavelengths (below). In accord with previous observations, UV-vis. spectra recorded for MPA showed no absorption over the
wavelength range of interest, $290 < \lambda$ / nm $< 370$ (see supplemental information).





**Figure 5: Solution phase spectra of MEK (black line) and PCO (blue line) obtained dilute in cyclohexane in this work, alongside gas-phase spectrum of MEK (red dashed line) recorded by (Martinez et al., 1992) and recommended by IUPAC. Calculated gas-phase**
**spectrum of MEK (orange dot-dashed line), calculated spectra of PCO for the gas-phase (light blue dot-dashed line) and the solution-phase (dotted dark blue line). The insert (top-right) displays Beer-Lambert plots obtained at 290 nm in this work, yielding a molar absorption coefficient $\varepsilon_{4(290\,nm)}$ = (20.6 ± 0.2) cm$^{-1}$M$^{-1}$ for PCO and $\varepsilon_{5(290\,nm)}$ = (16.4 ± 0.2) × 10$^{-20}$ cm$^2$ molecule$^{-1}$ for MEK.**

### 3.3 UV-vis. cross-sections σ(λ) as determined *in silico*

Computational photochemistry can be used to estimate photoabsorption cross-sections both in gas phase and within an implicit
solvent and determine what is the theoretical shift in wavelength and intensity caused by cyclohexane. The nuclear ensemble approach (NEA) (Crespo-Otero and Barbatti, 2012) was used to predict the photoabsorption cross-section for PCO and MEK (see Sec. 2.3 for the computational details), and we propose here a brief discussion of this technique as it is not commonly applied for the calculation of UV-vis. cross-sections. A common strategy to approximate a photoabsorption cross-section using quantum chemistry – the single-point approach – consists in locating the minimum-energy geometry in the ground electronic state for the molecule of interest, and to calculate for this geometry the vertical transitions and oscillator strengths towards the



excited electronic states. A photoabsorption cross-section can be obtained by broadening this single line with a Gaussian or Lorentzian function. The issue with the single-point approach is that it does not account properly for the quantum delocalisation of the ground-state nuclear wavefunction and non-Condon effects, that is, the fact that the transition dipole moment may vary with the nuclear configuration – as aspect particularly important for symmetry-forbidden transitions. The NEA offers a rather

straightforward strategy to calculate photoabsorption cross-sections that account for the broadening coming from quantum delocalisation and the possible enhancement in transition probability caused by non-Condon effects. We note that the NEA does not account for contributions from Franck-Condon factors, that is, overlaps of nuclear wavefunctions on the ground and excited electronic states, and as such does not reproduce the vibronic structure of a given electronic transition. We refer the interested reader to Prlj et al. (2022) – a recent work where the NEA is discussed in detail in the context of atmospheric

photochemistry.

The NEA photoabsorption cross-section calculated for MEK in gas phase shows a close agreement with the experimental cross-section in terms of intensity and centre of the $S_1 \leftarrow S_0$ band, which exhibits a $n\pi^*$ character. Including an implicit solvent (cyclohexane) leads to a slightly more intense cross-section predicted by the NEA for MEK, again mimicking closely the shift observed experimentally when comparing the gas-phase cross-section with the MEK cross-section measured in cyclohexane.

Comparing the MEK cross-section calculated in cyclohexane with that obtained for PCO shows that the latter is more intense and red-shifted, in line with the measured cross-section in cyclohexane. The difference in width between the experimental and theoretical cross-sections is caused by the approximations underlying the NEA (neglect of vibrational progressions). From a theoretical perspective, comparing the changes in photoabsorption cross-sections in gas phase *versus* cyclohexane for MEK and PCO, shows that the gas-phase photoabsorption cross-section is weakly red-shifted (0.0275 eV in average) with respect to

the cross-section in cyclohexane and its intensity is weakened by a factor 0.86.

**3.4 Estimation of photolysis rate coefficients (*j* values)**

Having measured the absorption cross section of PCO, the photolysis rate can then be estimated according to Equation 2:

$$j = \int \varphi(\lambda)\sigma(\lambda)F(\vartheta, \lambda)\, d\lambda \qquad\qquad\qquad (Eq.2)$$

Where $j$ is the first order photolysis rate coefficient, $\varphi$ stands for the quantum yield, $\sigma$ is the absorption cross section at

wavelength $\lambda$ and temperature $T$, and $F$ is the actinic flux at wavelength $\lambda$ and solar zenith angle $\theta$ (Fig. 7). As for many other organic compounds, quantum yields for PCO are unknown, and the MCM uses MEK as a surrogate with reasonably established photochemistry. Accordingly, the photolysis rate was calculated using $\varphi = 0.16$, the quantum yield determined by Pinho et al. (2005) for the wavelength range 275 to 380 nm. This value is different than that recommended IUPAC of $\varphi = 0.34$ (Atkinson et al., 2006), with the value from Pinho et al. (2005) optimised with environmental chamber data. The actinic flux used for the

estimation of $j$ values was obtained using the NCAR Tropospheric Ultraviolet and Visible (TUV) Radiation Model (Madronich and Flocke, 1997), with a solar zenith angle of 60° and an $O_3$ column of 350 DU (Dobson Units).



The integrated photolysis rate coefficient calculated via Eq.2 for PCO gives a value of $j_4 = 2.4 \times 10^{-6}$ s$^{-1}$ in the considered conditions, using $\sigma_4(\lambda)$ values from the UV-vis. spectrum recorded in cyclohexane (Fig. S6). The same procedure for the calculation of $j$ for MEK leads to a quite different result (Fig. 6), with $j_5 = 0.8 \times 10^{-6}$ s$^{-1}$ estimated using the cross sections

determined in this work (see Fig. S6) and $j_5 = 0.9 \times 10^{-6}$ s$^{-1}$, determined using cross sections from Martinez et al. (1992) (Fig. 6). This value is in reasonable agreement with the $j_5 = 1.3 \times 10^{-6}$ s$^{-1}$ calculated according to the MCM parameterisation at a similar zenith angle (Saunders et al., 2003a).

Figure 6: *j*-values for the photodissociation of PCO (R$_4$) and MEK (R$_5$). calculated via Eq.2. An actinic flux (the green dashed line), for conditions of $\vartheta = 60°$ and 350 DU from the NCAR (TUV) Radiation Model, together with $\varphi$ (275 - 380 nm) = 0.16 was used for all (Eq.2) *j*-value calculations. The blue striped area derives from use of $\sigma_4(\lambda)$ values recorded in experiments using cyclohexane solvent having applied the scaling factor of 0.95 to best reproduce gas-phase $\sigma_4(\lambda)$ values (see section 3.5), integrating to $j_4 = 2.3 \times 10^{-6}$ s$^{-1}$. Also displayed are *j*-values for photolysis of MEK (R$_5$) calculated using $\sigma_5(\lambda)$ from the gas-phase spectrum of Martinez et al. (1992) (red squares area, integrated to $j_5 = 0.9 \times 10^{-6}$ s$^{-1}$).



## 3.5 Discussion

The study of $R_1$ and $R_2$ via PLP-LIF to determine the rate coefficients of these reactions led to the values reported above with their (statistical only) standard error. Considering the systemic errors, we quote more realistic values of $k_1$ (296 K) = (1.2 ±
0.2) × 10⁻¹² cm³ molecule⁻¹ s⁻¹ and $k_2$(296 K) = (1.3 ± 0.3) × 10⁻¹² cm³ molecule⁻¹ s⁻¹, that take into account the error over the estimation of [VOC]. Both $k_1$ and $k_2$ are in good agreement with the literature data by Wallington and Kurylo (1987a) and by Wallington et al. (2001). Whilst this agreement was pleasing, all three studies used similar PLP methods and thus have a similar reliance on [VOC] measurements; confirmation of these results by e.g. the relative rate technique would be worthwhile in future. These results are also in agreement with the rate coefficients estimated via the Structure Activity Relationship (SAR)
method elaborated by Jenkin et al. (2018a) (Table 2). In addition, SAR product distribution for (R1) is in accordance with the one observed by (Wallington and Kurylo, 1987a) and recommended by the MCM (Master Chemical Mechanism)(Jenkin et al., 2003; Saunders et al., 2003a), which indicate (R1b) as the dominant route (Fig. 1, Table 2).

**Table 2 – Summary of $k_1$, $k_2$ and branching ratios.**

|  | **This work** | **Wallington et al.** | **MCM** | **SAR  (Jenkin et al., 2018b)** |
|---|---|---|---|---|
| $k_1$[a] | 1.2 ± 0.2 | 1.21 ± 0.05 [b] | 1.2 | 1.46 |
| %($k_{1a}$) | - | Minor | 0 | 9% |
| %($k_{1b}$) | - | Major | 100% | 91% |
| $k_2$[b] | 1.3 ± 0.3 | 1.20 ± 0.03 [b] | - | 1.34 |
| %($k_{2a}$) | - | 16% | - | 21% |
| %($k_{2b}$) | - | 84% | - | 79% |

**Notes: $k$ values in units of 10⁻¹² cm³ molecule⁻¹ s⁻¹. a) All $k$ values were recorded at ambient T b) Wallington and Kurylo (1987a). c) (Wallington et al., 2001). See SI for SAR calculation. d) $k_{1a}$, $k_{1b}$ and $k_{2a}$, $k_{2b}$ refers to Fig.1.**

However, as it is evident in Fig. 3 and Fig. 4, the temperature dependence of the SAR predictions is still unable to predict those observed experimentally for those two OVOC reactions. In the case of MPA, where results from this work together with literature data from Wallington et al. (2001) reveal a "U-shaped" non-Arrhenius $k(T)$ described by a 4-parameter expression, the SAR would appear to underestimate the relative importance of the conventional Arrhenius-like contribution to the overall $k(T)$, especially at $T > 300$ K. The SAR does appear to account for the complex anti-Arrhenius behaviour at lower temperatures, where direct H-abstraction is slow and the reaction proceeds (by analogy with similar OH + OVOC reactions) via formation of 6-member ring complexes featuring a hydrogen bond between the carbonyl function and the OH radical. Jenkin et al, (1998) do note that additional rate coefficient data would be highly valuable for further evaluation and constraining of the SARs for OVOC and multifunctional species.



For OH + PCO (R1), the experimental results have not provided enough evidence for any complex non-Arrhenius temperature dependence; experiments at lower temperature may yet bring further clarification. The $k(T)$ values from SAR so predict an anti-Arrhenius trend similar to the one calculated for MPA, somewhat different from the experimental results, although in
reasonable agreement in ambient temperature. Some caution should therefore be exercised when using the SAR to estimate product yields for (R1 – R2), even at $T \approx 298$ K, where the SAR appears to predict accurate $k$-values but potentially for the wrong reasons.

The lack of absorption cross-section and quantum yield data for many oxygenated organic molecules is an obstacle to the estimation of photolysis rates. Here, UV-Vis. spectra of PCO and MPA were recorded in the actinic light range ($\lambda > 290$ nm)
to help clarify the role of photolysis as an alternative (to OH reaction) removal path to oxidation. Cyclohexane was selected as a solvent for these solution-phase spectra (Nakashima et al., 1982); tests on the well-characterised molecule indicate that such solution-phase spectra can provide reasonable approximation of the gas-phase equivalent spectra, with a minor overestimation of the gas-phase values (see Fig. 5). Our UV-vis. results indicate a ratio of 0.95:1 between the integrated gas phase spectrum and the one recorded in solution (See SI for spectra integration). Computational photochemistry calculations
validated the experimental results, confirming an apparently systematic increase in photoabsorption when measuring the cross section in a cyclohexane solution. More precisely, a similar factor of 0.86:1 was calculated via NEA for the gas phase to solution phase spectra.

Assuming then that PCO has a similar behaviour to that observed for MEK, absorption cross sections recorded in cyclohexane solution may be used in photolysis rate determinations. In the absence of any quantum yield data for PCO, a value of $\varphi = 0.16$,
the quantum yield used for MEK by the MCM, was used here. However, $\sigma_4$(290 - 330 nm) were more intense than $\sigma_5$(290 - 330 nm) and the peak was significantly red-shifted (see Fig. 5), to wavelengths where light is abundant in the troposphere. Using the $\sigma_4$ photolysis rate determined in this work for PCO we estimated $j_4 = 2.4 \times 10^{-6}$ s$^{-1}$ (Fig. S6) and applying a scaling factor of 0.95 to $\sigma_4(\lambda)$, we estimated a very similar value of $j_4 = 2.3 \times 10^{-6}$ s$^{-1}$ (Fig. 6). Accordingly, PCO shows a significantly larger photolysis rate than the one estimated for MEK under similar atmospheric conditions (Fig. 6), with and $j_5 = 0.9 \times 10^{-6}$
s$^{-1}$. As a confirmation, the spectra recorded *in silico* for PCO and MEK describe a similar trend, as illustrated in Fig. 5.

Clearly the use of MEK as proxy for PCO in the MCM and other chemical models is not satisfactory and is likely to lead to an underestimation of the role of photolysis as regards PCO degradation. The lack of $\phi(\lambda)$ data means that a more accurate evaluation of the photolysis rate is not possible.

## 4 Atmospheric implications and conclusions.

The atmospheric chemistry of PCO and MPA has been investigated and compared to the limited number of previous studies. Both MPA and PCO react slowly with OH when compared to toluene, which reacts with OH about five times faster than either of these oxygenates (Mellouki et al., 2021).



From determinations of $k_1$(296 K and $k_2$(296 K) in this work, we have estimated lifetimes (Eq.3) for PCO and MPA with respect to reaction with OH in the troposphere:


$$\tau = \frac{1}{[OH] \times k_{OH}}$$ (Eq.3)

Using a mean tropospheric $[OH] = 1.13 \times 10^6$ molecule cm$^{-3}$ (Lelieveld et al., 2016b), the calculated lifetime with respect to OH is 9 days for PCO and 9 days for MPA. These lifetimes suggest a relatively low atmospheric reactivity, especially when compared to toluene ($\tau = 2$ days) and may allow for some dispersal or the primary emission prior to formation of ozone and other secondary pollutants. As discussed above, the UV-visible spectrum of MPA suggests that photolysis in the troposphere

($\lambda > 290$ nm) is not significant, and the main chemical loss process is therefore reaction with OH. By contrast, PCO absorbs light at 290 nm and above (Fig. 5) and hence (by analogy to the known photochemistry of other ketones) will incur photolysis losses. The photolysis rate estimated in this work ($j_4 = 2.3 \times 10^{-6}$ s$^{-1}$) using the scaled solution-phase cross section data and the quantum yield recommended by the MCM, produces a lifetime of 5 days with respect to photolysis ($\tau_p$) under the selected atmospheric conditions. Overall, the lifetime of PCO can be estimated using (Eq.4), shortening $\tau$ to 3 days (Table 3).


$$\frac{1}{\tau} = \frac{1}{\tau_p} + \frac{1}{\tau_{OH}}$$ (Eq.4)

Although this result strongly depends on the atmospheric conditions considered, and despite the acknowledged uncertainty on the estimation of $j$ because of lack of gas-phase absorption cross-section and quantum yield data, this shorter lifetime indicates how photolysis represents an important process for PCO and may outcompete reaction with OH as a removal pathway.

Photochemical parameters obtained in this work can further be used in chemical models and in calculations of estimated Photochemical Ozone Creation Potential (POCP$_E$) values for environmental assessments. The method was introduced by Jenkin et al. (2017) to take into account for size and structural features of the molecule, reactivity with OH and the presence of suitable chromophores. Here, POCP$_E$ values for North-Western European conditions were calculated for PCO and MPA (Table 3) using photochemical data determined in this work. The value of POCP$_E = 11$ for MPA may be quoted with some

confidence, given the good agreement in rate coefficients determined here and in previous work, and the absence of an active chromophore at relevant wavelengths. By contrast, three scenarios were considered for PCO, given the remaining uncertainties in the rate of (R$_4$). First, following the guidance in Jenkin et al. (2017), whereby one single value is recommended to account for photolysis of all aliphatic ketones, we calculated POCP$_E = 26$ for PCO. However, this photolysis parameter appears to be based upon cross-section and quantum yield values from the MCM, determined for photolysis of MEK (R$_5$). We cannot rule

out that the (unmeasured) quantum yields for PCO photolysis are small; in our second scenario we estimate that a consequently small value of POCP$_E = 12$ may be appropriate, ($\phi = 0$ so no photolysis, OH loss only). A third scenario would account for the enhanced rate of photolysis of PCO due to the more intense and red-shifted $\sigma_4(\lambda)$ determined in this work, and/or a higher quantum yield ($0.16 < \phi < 1$). However, a quantitative evaluation of this consequently enhanced POCP$_E > 26$ is beyond the scope of this work and would require detailed modelling studies.



**Table 3 – Lifetime and POCP$_E$ for PCO and MPA.**

|  | Lifetime, $\tau$ (days)[a] | | | POCP$_E$[b] | |
|---|---|---|---|---|---|
|  | $\tau_{OH}$ | $\tau_p$ | $\tau$ | POCP$_E$ (OH only) | POCP$_E$ (OH & photolysis) |
| **MPA** | 9 | - | 9 | 11 | 11 |
| **PCO** | 9 | 5 | 3 | 12 | 26[c] |

**Notes: [a] atmospheric lifetime ($\tau$) estimated using Eq. (4), based upon $k$(296 K) from this work and a value of [OH] = 1.13 × 10⁶ molecule cm⁻³ (Lelieveld et al., 2016a). [b] POCP$_E$ estimated according to Jenkin et al. (2017) method, see SI for more information. [c] does not take into account larger $j_4$ value and uses Jenkin et al. (2017) formula based on $\sigma_5(\lambda)$ and $\phi_5(\lambda)$.**

Comparing these POCP$_E$ values to traditional hydrocarbon solvents that MPA and PCO are meant to replace (Byrne et al., 2018a), these sustainable and non-toxic solvents show a much lower POCP$_E$. Toluene for instance has a POCP$_E$ of 45, which greatly exceed the POCP$_E$ scenarios estimated for PCO and MPA. Taken together, results from this work and elsewhere suggest that MPA has a moderate reactivity in the troposphere. PCO is likely more reactive due to photolysis (R$_4$). Even if significant uncertainties remain regarding PCO photochemistry, the use of both PCO and MPA would appear to have several advantages and to not adversely impact on air quality when compared to traditional harmful and non-sustainable solvents.

**Author contribution**

Laser-based experiments were designed by TJD and conducted by CM, ÚEH, JKD and TJD. UV-Vis experiments were conducted by CM, ATR and ARR. Quantum chemical calculations were carried out by BFEC and DH. The manuscript was written by CM, BFEC and TJD with assistance from other authors. TJD, FB and CRM conceived of the overall project.

**Competing interests**

The contact author has declared that none of the authors has any competing interests.

**Acknowledgements**

CM thanks the Dept. of Chemistry at York for a PhD scholarship. The authors thank Sacha Madronich (Atmospheric Chemistry Observations and Modeling Laboratory, National Center for Atmospheric Research, Boulder, CO, USA) for the reference actinic flux calculations. David Pugh is thanked for giving access to the UV-visible spectrometer. The authors thank the York Technical Support team, in particular Danny Shaw, Abby Mortimer, Mark Roper, Stuart Murray, and Chris Rhodes. Andrew



Rickard acknowledges support provided by the UK NERC National Centre for Atmospheric Science (NCAS) Air Pollution
Science Programme.

415



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
