# Peer review of "Atmospheric oxidation of new 'green' solvents part II: methyl pivalate and pinacolone"

_EGUsphere, 2023_

## Referee Comment (RC1)

**Atmospheric oxidation of new 'green' solvents part II: methyl pivalate and pinacolone**

**Major Comment**

This paper is well written and gets across its results in a digestible way.

The photolysis calculation using MEK as the proxy can be improved/investigated further by using more up to date data, see reference below. This has a significant impact as the more up to date photolysis study reduced the photolysis rate significantly, factor of 2-3. The older experiments did not look at the $T$ dependence and assumed it insignificant. It turns out that at the wavelength of interest > 300 nm photolysis has a significant $T$ dependence

Indication of the potential photolysis products (at 266 nm) can be ascertained by seeing if R + O2 ➔ OH occurs at low pressure. CH3CO + O2 is well known to do this reaction. Has this been checked?

While I understand the need to do the absorption measurements in solution, and then use theory to back-out the "gas-phase" values? I would maybe concentrate on the wavelengths of atmospheric interest (>300 nm). What difference is there between calculated and measured in this range. Is the % difference similar over the whole wavelength range? I'm not clear of the PCO cross-sections you have used in the photolysis calculations.

The abstract mentions exploring "(*CH3)3CC(O)CH3 photolysis (R4)*" but it is not really as you have assumed it behaves as MEK and scaled the absorption cross-sections appropriately. You have only really explored the absorption cross-sections (theory and expt) but not the photolysis, where you have assumed it behaves MEK-like.

Maybe it should be stated in abstract that methyl pivalate does not photolyse in the atmosphere and therefore its lifetime in the atmosphere is control by only reaction with OH.

**Specific comments**

**Line 24**
*UV-vis. spectroscopy experiments and computational calculations were used to explore (CH3)3CC(O)CH3 photolysis (R4).*
The absorption spectrum was determined in the solution phase and via theory translated to the gas-phase. Theory does not explore photolysis.

**Line 25**
*Absorption cross sections for (CH3)3CC(O)CH3, $\sigma 4(\lambda)$, in the actinic region were larger and the maximum was red-shifted compared to estimates used in current state-of-science models. As a consequence, we note that photolysis (R4) is likely the dominant pathway for removal of (CH3)3CC(O)CH3 from the troposphere.*
Red-shifted compared to what? I presume you mean MEK. You need to point out that MEK is being used as a proxy for PCO, where you are assuming that photolysis yields are the same except that you are scaling the absorption cross-sections.

**Line 81**
*is therefore concerning.* This seems a little overdramatic.

**Line 106**
*"log gas temperature"* While you estimate 15% error in the [OVOC] what do you estimate the accuracy of the temperature? The thermocouple will not be in the same place as where the laser beams overlap. Maybe also give typical gas flow rates for a given pressure.

**Line 119**
*"S(t) = S0 exp(-Bt)"* Is there any residual signal at long times, i.e. do you have to add a baseline to Equation 1.

**Line 119**
***2.2 Absorption cross sections, $\sigma(\lambda)$, via UV-vis. Experiments***

It is noted that you state that the spectrometer measures over the range 250 – 400 nm, but the longest wavelength reported is 337 nm. Can you state why this is the case. It could be that you have hit the minimum absorption that can be measured, but as you only report cross-sections I cannot tell if this is the case.

**Line 207**
*"Such non-Arrhenius behaviour may be indicative of a change of mechanism, with direct hydrogen abstraction dominating at high temperatures, whilst pathways via hydrogen bonded pre-reaction complexes play an increasingly important role at lower temperatures."*

I do not think you have to invoke a different mechanism to explain this behaviour. It can be explained by the change in the rate determining step, from capture of the complex at low temperature to abstraction at high temperature, i.e., from outer to inner transition-state control as *T* is increased.

**Line 263**

*"and as such does not reproduce the vibronic structure of a given electronic transition."* Is this related to problems in matching the experimental spectrum at long wavelength, > 300 nm? Does this mean the theory will not predict reliably the *T* dependence of the spectrum, which is often evident in the tail of the spectrum?

**Line 269**

*"predicted by the NEA for MEK, again mimicking closely the shift observed experimentally when comparing the gas-phase cross-section with the MEK"* While this is fair comment at the max of the spectrum, it is the values at in the actinic flux region that should also be considered. Below I have plot the MEK experimental and theoretical cross-sections (taken from the SI), where difference in the important 300-330 nm can reach a factor of ca. 5.

[Figure]

**Line 276**

*"**3.4 Estimation of photolysis rate coefficients (j values)**"* It is unclear what PCO cross-sections have been used in these calculations, theory or experimental? Please state.

**Line 282**

*"Accordingly, the photolysis rate was calculated using $\varphi = 0.16$, the quantum yield determined by Pinho et al. (2005)"* This reference is for isoprene degradation!

**Line 288**

*The same procedure for the calculation of j for MEK leads to a quite different result (Fig. 6), with j5 = 0.8 × 10-6 s -1 estimated using the cross sections determined in this work (see Fig. S6) and j5 = 0.9 × 10-6 s -1 290 , determined using cross sections from Martinez et al. (1992) (Fig. 6).*

A more recent study should also be considered:

DOI: 10.1039/B419160A (Paper) *Faraday Discuss.*, 2005, **130**, 73-88

**Photolysis of methylethyl, diethyl and methylvinyl ketones and their role in the atmospheric HO$_x$ budget**

M. Teresa Baeza Romero, Mark A. Blitz, Dwayne E. Heard, Michael J. Pilling, Ben Price, Paul W. Seakins* and Liming Wang

Where the temperature dependence of the photolysis was explored. It is state in this abstract that this reduces the photolysis rate by a factor of 2-3. This should be explored, especially when Table 3 has PCO's removal being dominated by photolysis.

**Line 308**
*"confirmation of these results by e.g. the relative rate technique would be worthwhile in future."* Yes, it is noted that the 266nm laser will be photolysing the PCO. In fact, it is noted that if PCO photolyses to CH3CO then at low pressures it can be tested via the well-known reaction CH3CO + O2 ➔ OH. Was this considered?

**Line 308 "*anti-Arrhenius*" It is more generally referred to as "non-Arrhenius"**

**Line 355**
*"Our UV-vis. results indicate a ratio of 0.95:1 between the integrated gas phase spectrum and the one recorded in solution (See SI for spectra integration)."* Not sure where this 0.95 comes from. Is it theory gas-phase vs theory solution or theory vs experimental gas phase. Also, note that it is difference in the actinic flux region that is most important

**Line 383**
*"However, this photolysis parameter appears to be based upon cross-section and quantum yield values from the MCM, determined for photolysis of MEK (R5)."* Should also check the impact from the above paper on MEK, where *T* is considered.

---

## Author Comment (AC1)

**Author Response to Referee Comments on 'Atmospheric oxidation of new 'green' solvents part II: methyl pivalate and pinacolone'**

We thank both reviewers for their time and attention and for the insightful and constructive comments on our research paper. The suggestions have all contributed to an improved manuscript.

**Response to RC1:**

Referee comments in *italic blue*; our response in **bold black**.

*This paper is well written and gets across its results in a digestible way. The photolysis calculation using MEK as the proxy can be improved/investigated further by using more up to date data, see reference below. This has a significant impact as the more up to date photolysis study reduced the photolysis rate significantly, factor of 2-3. The older experiments did not look at the T dependence and assumed it insignificant. It turns out that at the wavelength of interest > 300 nm photolysis has a significant T dependence Indication of the potential photolysis products (at 266 nm) can be ascertained by seeing if R + O2 � OH occurs at low pressure. CH3CO + O2 is well known to do this reaction. Has this been checked? While I understand the need to do the absorption measurements in solution, and then use theory to back-out the "gas-phase" values? I would maybe concentrate on the wavelengths of atmospheric interest (>300 nm). What difference is there between calculated and measured in this range. Is the % difference similar over the whole wavelength range? I'm not clear of the PCO cross-sections you have used in the photolysis calculations. The abstract mentions exploring "(CH3)3CC(O)CH3 photolysis (R4)" but it is not really as you have assumed it behaves as MEK and scaled the absorption cross-sections appropriately. You have only really explored the absorption cross-sections (theory and expt) but not the photolysis, where you have assumed it behaves MEK-like. Maybe it should be stated in abstract that methyl pivalate does not photolyse in the atmosphere and therefore its lifetime in the atmosphere is control by only reaction with OH.*

**Many thanks for these interesting and useful suggestions. Several are addressed in specific comments with corrections or clarifications below. In relation to the various points made around the temperature dependence of quantum yields, this is now discussed in the manuscript. However, our experiments and subsequent analysis (photolysis lifetimes and POCP estimates) were limited to ambient temperature – a detailed modelling study would be required to fully incorporate $\phi\,(\lambda, T)$.**

**Specific comments**

*Line 24 UV-vis. spectroscopy experiments and computational calculations were used to explore (CH3)3CC(O)CH3 photolysis (R4). The absorption spectrum was determined in the solution phase and via theory translated to the gas-phase. Theory does not explore photolysis.*

**We agree with the correction. Line 24 now reads "UV-vis. spectroscopy experiments and computational calculations were used to explore cross sections for (CH3)3CC(O)CH3 photolysis (R4)."**

*Line 25 Absorption cross sections for (CH3)3CC(O)CH3, $\sigma 4(\lambda)$, in the actinic region were larger and the maximum was red-shifted compared to estimates used in current state-of-science models. As*

*a consequence, we note that photolysis (R4) is likely the dominant pathway for removal of (CH3)3CC(O)CH3 from the troposphere. Red-shifted compared to what? I presume you mean MEK. You need to point out that MEK is being used as a proxy for PCO, where you are assuming that photolysis yields are the same except that you are scaling the absorption cross-sections.*

**Line 25 now reads "Absorption cross sections for $(CH_3)_3CC(O)CH_3$, $\sigma_4(\lambda)$, in the actinic region were larger and the maximum was red-shifted compared to estimates (MEK values) used in current state-of-science models."**

*Line 81 is therefore concerning. This seems a little overdramatic.*

**"The lack of photochemical data for degradation of PCO and MPA in the troposphere, and OVOCs in general, is therefore concerning." With 'concerning' we mean that it deserves the attention of the scientific community.**

*Line 106 "log gas temperature" While you estimate 15% error in the [OVOC] what do you estimate the accuracy of the temperature? The thermocouple will not be in the same place as where the laser beams overlap. Maybe also give typical gas flow rates for a given pressure.*

**Thank you for the comment, accuracy of the thermocouple was +/- 2 K (1% or less of the recorded temperatures). An example of the flow rate for the typical pressure was also given. Line 106 now reads "Briefly, output from four mass flow controllers was mixed prior to entering a 400 cm³ Pyrex reactor equipped with thermocouples (± 2 K) and capacitance manometers. Output from these calibrated analogue devices was digitised and passed to a PC to regulate and log gas temperature, pressure and flow rate (typical flow rate of 1000 sccm at 60 Torr)."**
**This paper provides a short description of the PLP-LIF method, but a more detailed description was given in Mapelli et al., 2022 (line 105), where we point out that during the experiment the thermocouple was readily translated in and out the photolysis region to check the temperature.**

*Line 119 "S(t) = S0 exp(-Bt)" Is there any residual signal at long times, i.e. do you have to add a baseline to Equation 1.*

**Baselines were measured and subtracted before exponential fitting.**

*Line 119 2.2 Absorption cross sections, $\sigma(\lambda)$, via UV-vis. Experiments It is noted that you state that the spectrometer measures over the range 250 – 400 nm, but the longest wavelength reported is 337 nm. Can you state why this is the case. It could be that you have hit the minimum absorption that can be measured, but as you only report cross-sections I cannot tell if this is the case.*

**No clear absorbance was observed beyond 337 nm - simply noise. To obtain quasi-vapour spectra we kept the concentration to a minimum and as a consequence the likely small absorbances beyond 337 nm were not observed.**

*Line 207 "Such non-Arrhenius behaviour may be indicative of a change of mechanism, with direct hydrogen abstraction dominating at high temperatures, whilst pathways via hydrogen bonded pre-reaction complexes play an increasingly important role at lower temperatures." I do not think you have to invoke a different mechanism to explain this behaviour. It can be explained by the change in the rate determining step, from capture of the complex at low temperature to abstraction at high temperature, i.e., from outer to inner transition-state control as T is increased.*

We agree with the Reviewer, line 207 now reads: "Such non-Arrhenius behaviour may be indicative of a change of rate-determining step within mechanism, with direct hydrogen abstraction dominating at high temperatures, whilst pathways via hydrogen bonded pre-reaction complexes play an increasingly important role at lower temperatures"

*Line 263 "and as such does not reproduce the vibronic structure of a given electronic transition." Is this related to problems in matching the experimental spectrum at long wavelength, > 300 nm? Does this mean the theory will not predict reliably the T dependence of the spectrum, which is often evident in the tail of the spectrum?*

The Reviewer is correct, and this is one of the limitations of the NEA employed here. The validity of the NEA for the prediction discussed by the Reviewer would need to be assessed. We stress that our calculations are used here solely to validate the expected shift between solution and gas-phase photoabsorption cross-sections. We note that, in general, the NEA can incorporate temperature effects on the spectra, and this strategy was used to study the temperature dependency of the UV/vis spectra of azobenzene (see Š. Sršeň, J. Sita, P. Slavíček, V. Ladányi, D. Heger, J. Chem. Theory Comput. 16 (2020) 6428–6438, doi: 10.1021/acs.jctc.0c00579).

However, whether NEA can capture the T dependence for molecules studied in the present paper remains an open questions.

*Line 269 "predicted by the NEA for MEK, again mimicking closely the shift observed experimentally when comparing the gas-phase cross-section with the MEK" While this is fair comment at the max of the spectrum, it is the values at in the actinic flux region that should also be considered. Below I have plot the MEK experimental and theoretical cross-sections (taken from the SI), where difference in the important 300-330 nm can reach a factor of ca. 5.*

The Reviewer is correct that our estimated overall solvent shift was calculated from the integrated intensity, and we assume it is not wavelength dependent. Because the NEA cannot correctly capture the tail of the spectrum, we did not attempt to account for a possible wavelength dependence of this shift. Given that the spectral peak is coming from a single electronic transition, we believe this assumption is not unreasonable. For MEK, as discussed above, the scaling factor for the whole peak (0.95) is comparable to the scaling factor for the actinic region (0.93) and thus we believe this is not a major source of error in our modelling.

Line 267 now contains: "We stress that the NEA calculations presented here are predominantly used to investigate the effect of a solvent (cyclohexane) on the photoabsorption cross-section of MEK and PCO."

*Line 276 "3.4 Estimation of photolysis rate coefficients (j values)" It is unclear what PCO cross-sections have been used in these calculations, theory or experimental? Please state.*

Thanks to the Reviewer for this correction, cross-sections for PCO were the ones measured in solution, scaled by 0.93 (In the previous version the scaling factor was 0.95 but it was recalculated for 300-330 nm). Line 293 now reads "The integrated photolysis rate coefficient calculated via Eq.2 for PCO gives a value of $j_4 = 2.4 \times 10^{-6}$ s$^{-1}$ in the considered conditions, using

$\sigma_4(\lambda)$ values from the UV-vis. spectrum recorded in cyclohexane, scaled by the sol-to-gas scaling factor 0.93, determined as described above."

*Line 282 "Accordingly, the photolysis rate was calculated using $\varphi = 0.16$, the quantum yield determined by Pinho et al. (2005)" This reference is for isoprene degradation!*

**The reference is correct. Pinho et al. studied MEK as a degradation product of butane in that paper, which is consequently cited by the MCM for MEK quantum yields.**

*Line 288 The same procedure for the calculation of j for MEK leads to a quite different result (Fig. 6), with j5 = 0.8 × 10-6 s -1 estimated using the cross sections determined in this work (see Fig. S6) and j5 = 0.9 × 10-6 s -1 290 , determined using cross sections from Martinez et al. (1992) (Fig. 6). A more recent study should also be considered: Where the temperature dependence of the photolysis was explored. It is state in this abstract that this reduces the photolysis rate by a factor of 2-3. This should be explored, especially when Table 3 has PCO's removal being dominated by photolysis.*

**Many thanks for providing the useful reference by Romero et al., this was included in the paper and the parameterized quantum yield was also considered. Our UV-vis. experiments were limited to room temperature and estimation of *j*, and the use of the parameterised quantum yield did not bring to a lower value. However, it was noted in the atmospheric implications that T dependence should also be considered. A new table (Table 2) was introduced in the text to summarise the *j*-values obtained using different quantum yields and cross-sections.**

**Line 286 now reads: "Subsequently Romero et al. (2005) reported quantum yields lower than the one estimated by Raber and Moortgat (1995), and their work stress out the importance of investigating the temperature dependence and the spectral distribution of quantum yields to avoid overestimation.**

**Line 297 now reads: "Photolysis rates were also calculated using the parameterized $\varphi$ ($\lambda$, [*M*], *T*) formulated by Romero et al. (2005)". Using the parameterised quantum yield leads to fairly similar results, with $\lambda$-dependent values larger than MCM-values by a factor ~ 1.2. Table 2 summarises the results obtained from all the available cross-sections and quantum yields.**

*Line 308 "confirmation of these results by e.g. the relative rate technique would be worthwhile in future." Yes, it is noted that the 266nm laser will be photolysing the PCO. In fact, it is noted that if PCO photolyses to CH3CO then at low pressures it can be tested via the well-known reaction CH3CO + O2 ⌈ OH. Was this considered?*

**Under the typical conditions of our experiment (60 Torr, 296 K) we did not detect OH in the absence of precursor (H2O2) indicating that we were not sensitive to PCO photolysis.**

*Line 308 "anti-Arrhenius" It is more generally referred to as "non-Arrhenius"*

**We agree and the text was changed accordingly.**

*Line 355 "Our UV-vis. results indicate a ratio of 0.95:1 between the integrated gas phase spectrum and the one recorded in solution (See SI for spectra integration)." Not sure where this 0.95 comes from. Is it theory gas-phase vs theory solution or theory vs experimental gas phase. Also, note that it is difference in the actinic flux region that is most important*

The scaling factor was estimated from the ratio between the integrated area of the experimental gas phase (Martinez et al.) and the experimental solution (this work). We agree that the actinic flux region is most important, and we re-calculated the scaling factor only considering that spectral range (0.93 instead of 0.95). Line 345 now reads: "Our UV-vis. results indicate a ratio of 0.93:1 between the integrated gas phase spectrum by Martinez et al. (1992) and the one recorded in solution in this work (See SI) in the spectral range 300-330 nm."

*Line 383 "However, this photolysis parameter appears to be based upon cross-section and quantum yield values from the MCM, determined for photolysis of MEK (R5)." Should also check the impact from the above paper on MEK, where T is considered.*

We agree with the comment. Line 392 now reads: "T dependence of $\phi$ should also be considered and investigated, and considering only room temperature $\phi$ may lead to overestimation as pointed out by Romero et al. (2005)".

**Response to RC2:**

*This paper presents an experimental and numerical study of the atmospheric oxidation of methyl pivalate (MPA) and pinacolone (PCO), both considered as "green" solvents. First, the reactions rate constants of MPA and PCO with OH radicals are measured using the PLP-LIF (Pulsed Laser Photolysis – Laser Induced Fluorescence) technique in the temperature range (295-485 K), showing results in line with the scarce data from the literature. UV-Vis absorption cross sections of PCO were also obtained experimentally (in liquid phase) and numerically, and compared to data for MEK (Methyl Ethyl Ketone). The authors also propose an estimation of the photolysis rate coefficients for PCO. Finally, lifetimes for removal of PCO and MPA from the troposphere are calculated, suggesting relatively low reactivity of MPA and more reactivity for PCO due to photolysis.*

*In general, this paper is of good quality, describing a very comprehensive work combining experiments and modelling, even if it is not perfect and leaves open questions, notably because of the lack of comparable gas phase absorption cross sections and quantum yields data for PCO and MPA, the authors must have made approximations.*

*Minor considerations:*

*- As mentioned by the authors (line 107): "rate coefficient determinations in this work relied critically on accurate knowledge of [OVOC], here estimated to a precision of ±15%"*

*Indeed, it is often difficult to accurately determine the concentrations of OVOCs in this type of experiment. It is not clear how the mixtures of different concentrations of MPA and PCO diluted in nitrogen are prepared. It is therefore difficult to understand how the ± 15% uncertainty is estimated. Could the authors give more details on this aspect?*

**The OVOCs used in the experiments were supplied via a glass bulb that was then connected to an MFC and to the PLP-LIF apparatus. The bulbs were prepared at the Schlenk line using a standard method and their concentrations were determined through manometric measurements. The pressure in the line was monitored with two capacitance manometers, 10 Torr and 1000 Torr. A few millilitres of liquid OVOC were transferred into a glass finger and subjected to at least three cycles of freeze (77 K)-pump-thaw purification. The desired amount of OVOC was transferred into the bulb and filled with N$_2$ up to about 1000 Torr.**

*- Concerning LIF measurements (line 116): "This tuneable laser light was used to pump the Q11 transition of A$^2$S$^+$(v = 1) - X$^2$P(v = 0) at 281.997 nm for direct, off-resonant LIF detection of OH." What does this mean? At which wavelengths are resonance and non-resonance measurements made? The Q1(1) line of OH is not an isolated line (the R2(3) line is very close). Does this pose a problem for the interpretation of the LIF results? What is the wavelength resolution?*

**Following off-resonance excitation at 281.997 nm, all fluorescence that can pass through the 308 nm interference filter is collected by the PMT. A large proportion of this fluorescence is therefore likely on-resonance 308 nm radiation. The identity of the rovibrational excitation line is therefore not important.**

*- line 304: "Considering the systemic errors, we quote more realistic values of $k_1$ (296 K) = (1.2 ± 0.2) × $10^{-12}$ cm³ molecule$^{-1}$ s$^{-1}$ and $k_2$(296 K) = (1.3 ± 0.3) × $10^{-12}$ cm³ molecule$^{-1}$ s$^{-1}$, that take into account the error over the estimation of [VOC]". It is not clear how this concentration error is taken into account in the final uncertainty calculation*

**The statistical uncertainty in the ambient temperature data (for instance $k_1$ (296 K) = (1.23 ± 0.09) × $10^{-12}$ cm³ molecule$^{-1}$ s$^{-1}$) was combined with the estimated 15% uncertainty in concentration via a squared sum. However, the error relative to $k_2$, $k_2$(296 K) = (1.3 ± 0.3) × $10^{-12}$ cm³ molecule$^{-1}$ s$^{-1}$ was reported by mistake and was corrected to $k_2$(296 K) = (1.3 ± 0.2) × $10^{-12}$ cm³ molecule$^{-1}$ s$^{-1}$.**

*- Line 177: "Similar experiments were conducted at different temperatures, pressures and in the presence / absence of $O_2$". The authors should clarify the value of such measurements (with and without $O_2$) and at least give an indication of the comparative results*

**In Table 1, the experiments where the bath gas was $N_2$ have a suffix label 'a' next to the pressure values. Experiments with $O_2$ as bath gas were labelled with 'b' (see table caption). We apologise as in the previous version there was an error with this labelling system but now it's correct.**

Martinez, R. D., Buitrago, A. A., Howell, N. W., Hearn, C. H., and Joens, J. A.: The near U.V. absorption spectra of several aliphatic aldehydes and ketones at 300 K, Atmos. Environ. Part A. Gen. Top., 26, 785-792, https://doi.org/10.1016/0960-1686(92)90238-G, 1992.

Raber, W. H. and Moortgat, G. K.: Photooxidation of selected carbonyl compounds in air: methyl ethyl ketone, methyl vinyl ketone, methacrolein and methylglyoxal, in: Progress and Problems in Atmospheric Chemistry, 318-373, 10.1142/9789812831712_0009, 1995.

Romero, M. T., Blitz, M. A., Heard, D. E., Pilling, M. J., Price, B., Seakins, P. W., and Wang, L.: Photolysis of methylethyl, diethyl and methylvinyl ketones and their role in the atmospheric HOx budget, Faraday Discuss., 130, 73-88; discussion 125-151, 519-124, 10.1039/b419160a, 2005.